# Efficacy of Liver Chemoembolization after Prior Cetuximab Monotherapy in Patients with Metastatic Colorectal Cancer

**DOI:** 10.3390/cancers15020541

**Published:** 2023-01-16

**Authors:** Marcin Szemitko, Elzbieta Golubinska-Szemitko, Jerzy Sienko, Aleksander Falkowski, Ireneusz Wiernicki

**Affiliations:** 1Department of Interventional Radiology, Pomeranian Medical University, 70-111 Szczecin, Poland; 2Department of General and Dental Diagnostic Imaging, Pomeranian Medical University, 70-111 Szczecin, Poland; 3Department of General and Transplant Surgery, Pomeranian Medical University, 70-111 Szczecin, Poland; 4Department of Vascular Surgery, General Surgery and Angiology, Pomeranian Medical University, 70-111 Szczecin, Poland

**Keywords:** colorectal cancer, metastases, TACE, irinotecan, cetuximab

## Abstract

**Simple Summary:**

The aim of the study was to investigate the efficacy of irinotecan-releasing beads in the treatment of metastatic colorectal cancer after prior cetuximab monotherapy in patients with metastatic colorectal cancer, as it has been suggested that the development of resistance to anti-EGFR antibodies may result in resistance to irinotecan. We found no statistically significant difference in radiological response to TACE treatment according to whether cetuximab therapy was previously used or not, but our study showed a significant correlation between low baseline CEA values and response to treatment, which may favor this group of patients in qualifying for TACE treatment.

**Abstract:**

**Purpose**: Chemoembolization of liver lesions, metastatic from colorectal cancer (CRC), with irinotecan-loaded microspheres shows less efficacy if applied after previous systemic chemotherapy. This is because cancer cells acquire resistance to previously used chemotherapeutic agents, e.g., irinotecan or perhaps via, e.g., modulations of EGFR receptors after use of anti-EGFR antibodies. **Objective:** To evaluate the effects of prior treatment with anti-EGFR (cetuximab) antibodies on the efficacy of chemoembolization, with irinotecan-loaded microspheres, of liver lesions metastatic from CRC. **Patients and methods**: The study included 50 patients (27 female, 23 male) with inoperable liver metastases in the course of CRC who underwent a total of 192 chemoembolization procedures with microspheres loaded with 100 mg of irinotecan. Chemoembolization of the right or left liver lobes was performed alternately at three-week intervals. Patients were divided into two groups: group A (*n* = 26): patients who had previously received anti-EGFR (cetuximab) antibodies; and group B (*n* = 24): patients who had never received anti-EGFR antibodies. Response to treatment was assessed according to mRECIST criteria. Overall survival time (OS) was calculated using the Kaplan–Meier method. Evaluation of adverse effects was performed according to the Cancer Therapy Evaluation Program Common Terminology Criteria for Adverse Events (Version 5.0). **Results**: Analysis did not show a statistically significant difference in radiological response between the two groups: partial response: 36.2% in group A and 32.9% in group B (*p* = 0.139); and stable disease: 19.2% in group A and 21.7% in group B (*p* = 0.224). Post-treatment progression was comparable at 46.2% in group A and 41.6% in group B (*p* = 0.343). There was a significant difference in OS (*p* = 0.043 log-rank test), however, prior treatment with cetuximab showed no significant effect on OS in a Cox proportional hazards regression model HR 1.906 (0.977–3.716), *p* = 0.058. Mean OS was 15.2 months (95% confidence interval (Cl): 6 to 23 months) in group A and 13.1 months (95% Cl: 7 to 22 months) in group B. In both groups, there was a negative correlation between carcinoembryonic antigen (CEA) levels below 10 mg/mL before surgery and OS (hazard ratio (HR) 0.83 (0.47–8.43), *p* = 0.005 in group A and HR 1.02 (0.56–7.39), *p* = 0.003 in group B). There was no significant difference in the number of prominent complications between group A (7 complications) and group B (6 complications), *p* = 0.663. **Conclusions**: Previous therapy with anti-EGFR antibodies before treatment with irinotecan chemoembolization of liver metastatic lesions did not have a significant effect on radiological response to treatment or post-treatment progression. However, higher baseline levels of CEA (>10 ng/mL) were correlated with worse OS (*p* = 0.039).

## 1. Introduction

Liver metastases occur in a majority of patients with colorectal cancer [1,2]. Unfortunately, surgical resection is only possible in about 10–15% of patients [3,4]. For other patients, the standard treatment option is palliative systemic chemotherapy, most often with the use of 5-fluorouracil and leucovorin in combination with irinotecan or oxaliplatin [5]. In cases where the liver is the only, or predominant, site of metastasis, it is possible to use intra-arterial chemoembolization with irinotecan-loaded microspheres (TACE), which in some studies has been shown to be effective [6,7].

A breakthrough that increased the effectiveness of systemic chemotherapy was the use of antibodies against the epithelial growth factor receptor (EGFR). In particular, combination therapy of anti-EGFR with FOLFIRI (5-fluorouracil, leucovorin, and irinotecan) or FOLOFOX (5-fluorouracil, leucovorin, and oxaliplatin) has shown greater or comparable efficacy to transarterial chemoembolization (TACE) as a first line of treatment [8,9]. This has resulted increasingly in the use of TACE only in the third or fourth lines of treatment, when, due to the development of tumor cell resistance, the number of possible chemotherapeutic agents decreases significantly [10,11]. Available results suggest that TACE is less effective when used after failure of prior systemic chemotherapy [12]. However, it should be noted there are clear disparities between studies resulting from differences in the qualification of patients and previous systemic chemotherapy regimens [13].

One suggested mechanism by which cancer cells can acquire resistance to irinotecan is an increase in the expression of the EGF receptor [14], which is also a target for anti-EGFR antibodies. It has been suggested that the development of resistance to anti-EGFR antibodies causes resistance to irinotecan and hence reduces the efficacy of TACE in later lines of treatment [15]. In our study, the efficacy of chemoembolization with irinotecan-loaded microspheres (TACE) of liver metastatic lesions was analyzed in relation to previous treatment with anti-EGFR monoclonal antibodies (cetuximab), with the aim of identifying the group of patients in whom chemoembolization may be most beneficial.

## 2. Materials and Methods

This retrospective study evaluated the results of chemoembolization procedures for unresectable liver metastatic lesions in the course of CRC, performed between July 2017 and March 2021. The Bioethics Committee of the Pomeranian Medical University in Szczecin approved this study.

The analysis included 50 patients (27 women and 23 men) with progression of metastatic lesions after previous palliative chemotherapies. All patients received first-line palliative chemotherapy with irinotecan (FOLFIRI). In a second line, patients received chemotherapy with oxaliplatin (FOLFOX). After failure of these lines of chemotherapy and after excluding mutations in the *KRAS* and *BRAF* genes, some patients were qualified for monotherapy with anti-EGFR antibodies (cetuximab; patients included in group A). An intravenous loading dose of 400 mg/m^2^ of cetuximab (body surface area) was administered on day 1 of treatment, followed by an infusion of 250 mg/m^2^ (body surface area) administered once weekly.

Patients were divided into two groups: Group A (*n* = 26) in which the patients had received anti-EGFR antibody (cetuximab) treatment and Group B (*n* = 24) in which patients had not been treated with anti-EGFR antibodies due to the presence of *KRAS* or *BRAF* mutations. Using microspheres loaded with the cytostatic irinotecan (100 mg), a total of 192 chemoembolization procedures were performed

After consultation with a specialist oncologist, qualification for procedures was performed according to the recommendations of the European Society of Medical Oncology (ESMO). All patients previously underwent computed tomography (CT) or magnetic resonance imaging (MRI) of the abdominal and laboratory testing. Indications for treatment were the presence of CRC liver metastases unsuitable for resection or ablation, with progression after previous chemotherapy and age over 18 years old.

Exclusion criteria for the study were: involvement of more than 50% of liver parenchyma, ECOG > 2, ascites, bilirubin > 3 mg/dL, creatinine > 2 mg/dL, thrombocytopenia < 50,000/mcl, and allergic reaction to contrast in the past. Response to treatment was assessed by CT scan according to the modified Response Evaluation Criteria in Solid Tumors (mRECIST) criteria.

The treatment regimen consisted of four treatments or two if only one liver lobe was involved. Alternating embolization of branches of the right or left hepatic artery and additional arteries supplying the liver lesions were performed with three-week intervals between treatments. Microspheres (Embozene Tandem 100 µm; CeloNova Biosciences, now Varian Medical System, Inc, Palo Alto, CA, USA) were used. After loading irinotecan onto the microspheres, the supernatant was removed from the syringe and the microspheres were mixed with 10 mL of contrast agent (Iodixanolum 320 mg I/mL).

The procedures were performed by interventional radiologists with certified skills in interventional radiology.

On the day before and the day of the procedure, each patient received steroids (Dexamethasone), proton pump inhibitors (Omeprazolum), an antiemetic drug (Ondansetron), and prophylactic antibiotics (Cefazolin), and an infusion of 1000 mL of 0.9% NaCl.

### 2.1. Procedure

The puncture of the right or left common femoral artery was performed using the Seldinger method. The celiac trunk (or superior mesenteric artery in the case of an anatomical variant) was catheterized using a SIM 5F catheter (Cordis, Miami Lakes, FL, USA). Vascularization of the liver and metastatic lesions was evaluated in arteriography and cone-beam CT.

Each administration of embolizate was preceded by an injection of 1–2 mL of lidocaine into the microcatheter (Progreat^®^ 2.7F micro catheter, Terumo, Tokyo, Japan). The mixture of microspheres and contrast agent was slowly administered (at a rate of approximately 1 mL/min) under fluoroscopy. Microsphere administration was continued until “near-stasis” (a stasis that resolves within seconds) was achieved at the level of the vessels supplying the tumors.

Pain that occurred during and after the surgery was controlled with intravenous morphine infusion. Ondansetron 8 mg i.v., dexamethasone 8 mg i.v., and cefazolin 1 g i.v. were administered prophylactically twice daily. Most patients were discharged from the hospital within 24 h after surgery.

According to the standards of the Cancer Therapy Evaluation Program Common Terminology Criteria for Adverse Events (Version 5.0) complications were assessed on the basis of examinations of the patient during hospitalization and follow-up visits. Data were recorded for statistical evaluation (Excel 2007; Microsoft, Washington, DC, USA).

### 2.2. Feasibility of Chemoembolization

The 50 patients included in the study underwent a total of 192 chemoembolization procedures. In 46 patients with two lobes involved, 184 chemoembolization procedures were performed. A total of 4 patients with unilobar involvement underwent 8 chemoembolization procedures. The technical success rate of the treatments was 100%.

### 2.3. Imaging and Tumor Response

Before and one month after the last procedure, imaging was performed using multi-phase computed tomography or contrast-enhanced magnetic resonance imaging to assess response according the mRECIST criteria.

### 2.4. Statistical Analyses

Continuous variables were given as arithmetic means and standard deviations or as medians and ranges. Qualitative variables were analyzed using χ2 tests. Continuous variables were compared using *t*-tests or Mann–Whitney U tests for variables with non-normal distributions. All above tests were performed using commercially available software (Statistica version 13.1. (StatSoft Polska, Krakow, Poland). Cumulative survival rates (OS) were expressed using Kaplan–Meier analysis from the date of a TACE patient’s first treatment to the date of that patient’s last follow-up visit or patient death. The risk factors of death were analyzed using univariate Cox proportional hazards models with 95% confidence intervals. Survival analyses were performed using IBM SPSS Statistics for Macintosh, version 29.0. (IBM Corp., Armonk, NY, USA). Significance of all tests was determined at the *p* = 0.05 level.

## 3. Results

### 3.1. Baseline Characteristics

Patient characteristics showed no differences between groups (Table 1).

### 3.2. Response

There was no statistically significant difference (*p* = 0.139) in partial response (PR) between the groups, with 9 patients (34.6%) in group A and 8 patients (33.3%) in group B. Stabilization of SD lesions occurred in 5 patients in group A (19.2%) and 5 from group B (20.8%) (*p* = 0.224). In contrast, progressive disease (PD) occurred in 12 patients from group A (46.2%) and 10 patients from group B (41.7%) (*p* = 0.343). One patient (4.2%) in group B, with a single metastatic lesion, had complete remission. (Figure 1).

### 3.3. Survival Analysis

Overall survival time in group A was significantly longer compared with group B (*p* = 0.043). The median survival time was 15.2 months (95% Cl: 6–23) in group A and 13.1 months (95% Cl: 7–22) in group B (Figure 2).

Univariate Cox’s regression model revealed that ECOG performance status 0 (*p* < 0.001) and less than 25% liver involvement (*p* = 0.013) was associated with better OS, whereas a high level of CEA (>10 ng/mL) was correlated with worse OS (*p* = 0.039). Previous cetuximab treatment showed no significant impact on OS (*p* = 0.058).

In multivariate analysis ECOG performance status (*p* < 0.003), the degree of liver involvement (*p* = 0.011) and CEA level (*p* = 0.043) before chemoembolization were found to have a significant effect on OS (Table 2).

### 3.4. Adverse Events

In the chemoembolizations performed, there were a total of 15 (7.8%) significant complications, 8 in group A and 7 in group B (*p* = 0.663). The type and number of complications are shown in Table 3. There were no deaths within 30 days after the procedure.

## 4. Discussion

Chemoembolization of liver metastases from CRC using irinotecan-loaded TACE microspheres is indicated when the liver is the sole or predominant site of metastasis. This allows limiting the frequency of irinotecan side-effects by reducing systemic exposure and delivery of a high dose of chemotherapeutic agent directly to the metastatic lesions [16]. Irinotecan is a semi-synthetic analog of camptothecin which is metabolized in the liver parenchyma by carboxylesterases (CES-1 and CES-2) into the active metabolite 7-ethyl-10-hydroxy-camptothecin (SN-38). The SN-38 inhibits DNA transcription several hundred times greater than that of irinotecan alone. Most SN-38 is produced in the liver parenchyma, from where it diffuses into tumor cells [17]. The mechanisms by which irinotecan resistance is acquired are not completely understood; some suggestions being a possible increased expression of EGFR receptors [18,19] and/or active efflux giving reduced intracellular accumulation of the drug [20]. Even less is known about the resistance of tumor cells to chemoembolization, where irinotecan has very different pharmacokinetic conditions. TACE embolization contributes to reduced drug washout as well as more efficient conversion to and release of the active metabolite irinotecan SN-38 in the liver. Moreover, post-embolization hypoxia lowers the tumor tissue pH, which enhances the conversion of irinotecan to its metabolite SN-38 in hepatocytes and increases its activity [21].

The use of anti-EGFR antibodies in the treatment of metastatic CRC, first used in the 3rd line, and then as part of combination treatments in 1st and 2nd line systemic chemotherapies, has clearly improved efficacy [22]. However, the efficacy of anti-EGFR antibodies is clearly dependent on the absence of *KRAS* and *BRAF* protooncogene mutations [23]. A *KRAS* mutation is found in tumor cells in about 40% of patients, and around 10% of patients have a *BRAS* mutation. During the course of treatment, anti-EGFR antibody resistance develops in more than 80% of cases, with the most commonly suggested mechanism being EGFR ligand overexpression [24,25]. Using an EGFR inhibitor in addition to SN-38 may possibly defeat resistance by increasing tumor cell apoptosis [26] and one study has confirmed the efficacy of chemoembolization with irinotecan in combination with anti-EGFR antibody therapy (cetuximab) [27]. Given that the EGFR receptor pathway possibly plays an important role in the acquisition of tumor cell resistance to both anti-EGFR antibodies and irinotecan, the use of both drugs together might have a reciprocal effect on the accumulation of resistance. Previous studies on the chemoembolization of CRC metastases have not analyzed the impact of possible tumor cell resistance resulting from previous cetuximab therapy. The relationships between previous anti-EGRF antibody therapy and the efficacy of TACE in the later stages of treatment have also not been investigated.

In the present study, the percentage of positive responses (PR + SD) to TACE in the 4th line of mCRC treatment was 55.4% in patients previously treated with cetuximab and 54.6% in those who were not. We found no statistically significant difference in radiological response to treatment according to whether anti-EGRF antibody therapy was previously used or not. This confirms the possible benefit of qualifying patients for TACE regardless of previous anti-EGFR antibody therapy.

However, we have demonstrated a possible significant difference in overall survival time, with a benefit for patients treated sooner with anti-EGFR antibodies, which were used only after KRAS and BRAS mutations were excluded. There are conflicting reports in the available literature regarding the impact of these mutations on patient survival [22,28]. In addition, our study showed a significant correlation between low baseline CEA values and response to treatment, which may favor this group of patients in qualifying for TACE treatment.

## 5. Conclusions

Previous therapy with anti-EGRF antibodies in patients treated with irinotecan chemoembolization of liver metastatic lesions does not show a significant effect on overall assessed responses to treatment. However, longer overall survival times were demonstrated for patients previously treated with cetuximab as well as with patients with low baseline carcinoembryonic antigen levels.

## 6. Limitations

The study was retrospective, involved patients from a single clinical center, and was non-randomized. Increasing the numbers of patients in each group would be advisable.

## Figures and Tables

**Figure 1 cancers-15-00541-f001:**
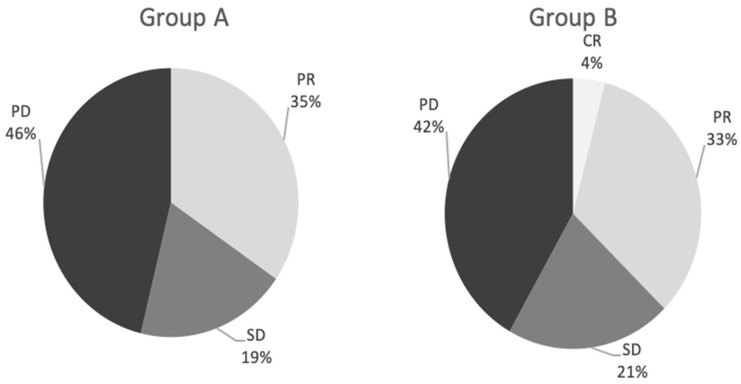
Response to treatment between the two groups.

**Figure 2 cancers-15-00541-f002:**
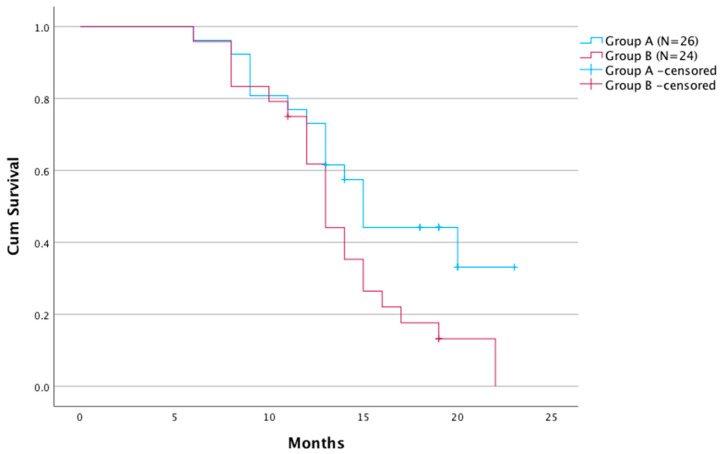
Kaplan–Meier survival analysis for the two groups.

**Table 1 cancers-15-00541-t001:** Patient characteristics. Comparison between the two groups was assessed by *t* or Chi-squared tests. *p* value < 0.05 was considered significant.

Parameter	Group A (*n* = 26)	Group B (*n* = 24)	*p*-Value
Age, median (range)	65.3 (32–74)	66.5(38–77)	0.426
Gender, female/male (*n*)	15/11	12/12	0.667
**ECOG status (*n*):**			0.323
0	10	8	-
1	12	13	-
2	4	3	-
**Tumor location (*n*):**			0.178
Bilobar	24	22	
Unilobar	2	2	
Number of liver metastases, median (range)	4.4 (1–10)	4.1(1–9)	0.139
Largest nodule size diameter, cm (median)	9.8	8.9	0.297
Extent of liver involvement (*n*, <25%/>25%)	21/5	19/5	0.401
Extrahepatic metastasis (*n*, %)	8	8	0.278
**Site of primary tumor (*n*):**			0.409
Left colon	15	14	
Right colon	11	10	
Prior liver surgery/ablation (*n*)	5/0	4/0	0.502
Prior locoregional therapy (*n*)	0	0	-
**TACE procedure performed for patient (*n*):**			0.178
4 procedures	24	22	
4 procedures	2	2	
**CEA level (*n*):**			
<10 ng/mL	12	11	0.578
>10 ng/mL	14	13	0.451
**CRC somatic mutation (*n*)**			
*KRAS* (Exon2)	-	21	
*KRAS*(non-Exon2)	-	1	
*BRAS* (V600E)	-	2	

**Table 2 cancers-15-00541-t002:** Cox regression hazard ratios (HR) in univariate and multivariate analysis for prediction of death.

Factor	Univariate Cox’s Regression HR (95% Cl) *p*-Value	Multivariate Cox’s Regression HR (95% Cl) *p*-Value
Age (>65 vs. ≤65)	2.760 (0.371–20.50), *p* = 0.321	
Gender (female vs. male)	1.959 (0.262–14.662), *p* = 0.512	
**ECOG status: (0 vs. 1 and 2)**	**0.155 (0.057–0.421), *p* <0.001**	**0.108 (0.024–0.477), *p* <0.003**
Largest nodule size diameter (<5 cm vs. >5 cm)	1.846 (0.821–4.232), *p* = 0.136	
**Extent of liver involvement (<25%/>25%)**	**0.375 (0.173–0.816), *p* = 0.013**	**0.185 (0.051–0.676), *p* = 0.011**
Previous cetuximab (yes vs. no)	1.906 (0.977–3.716), *p* = 0.058	
**CEA (>10 ng/mL vs. < 10 ng/mL)**	**2.374 (1.043–5.406), *p* = 0.039**	**3.330 (1.036–10.702), *p* = 0.043**
Extrahepatic metastasis (yes vs. no)	0.769 (0.090–6.600), *p* = 0.811	
Primary tumor resection (yes vs. no)	1.485 (0.674–3.271), *p* = 0.32	
Site of primary tumor (left colon vs. right)	1.452 (0.573–3.495), *p* = 0.452	
TACE procedure performed for patient (4 vs. 2)	6.132 (0.799–47.053), *p* = 0.081	

ECOG = Eastern Cooperative Oncology Group performance status; CEA = carcinoembryonic antigen.

**Table 3 cancers-15-00541-t003:** Number of complications in each group.

Adverse Event	Group A	Group B
Liver failure/ascites	2	1
Inflammation of the gallbladder	2	1
Occlusion of the main branch of the hepatic artery	0	2
Leukopenia < 2000/mm^3^	2	2
Liver abscesses	0	1
Anaphylactic reaction	2	0

## Data Availability

The data presented in this study are available on request from the corresponding author.

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
