# Peer review of "Efficacy of Liver Chemoembolization after Prior Cetuximab Monotherapy in Patients with Metastatic Colorectal Cancer"

_cancers, 2023, doi:10.3390/cancers15020541_

Round 1

Reviewer 1 Report

Szemitko et al. presented a research that focusing on evaluating the effects of prior treatment of cetuximab on chemoembolization and survival. Overall it is an interesting manuscript and demonstrate a clear conclusion.

The main concerns are:

1. Although the categories of the samples are balanced, the sample size is relatively small. The p-value of the survival testing is p=0.043 is almost close to the threshold 0.05. 

2. Table 2 shows the cox regression HR for predicting death. How did the authors differentiate the factors outside from the prior treatment of cetuximab. Is there any possibility that the nodule size diameter is the determinant of the survival rather than the prior treatment? As there are other factors are significantly different between those two groups, how to differentiate which factor is the determining one.

3. The title is a little bit of confusing, should at least include the test and the conclusion.

4. Were the patients sequenced for mutation analysis? Is there any driver mutation recognized or could contribute to the survival difference between group A and B?  

Minors:

1. The figure 1 seems to not be included in any of the manuscript. Why is it in the introduction? What is it for?

2. The figure 2 might be better remake into a pie chart.

3. Figure 3 is confusing. What is censored? It is too vague, need to at least be a 300 DPI figure and also need to include the p-value and sample size.  

Author Response

Dear reviewer, thank you for your suggestions. The main concerns are: 1. Although the categories of the samples are balanced, the sample size is relatively small. The p-value of the survival testing is p=0.043 is almost close to the threshold 0.05. We agree, the correlation is close to the significance level of 0.05, and the study group could be larger. However, this is a single-center study with patients who have progression of disease after second and third lines of chemotherapy. Due to advanced disease and/or patients’ overall general coniditon, a large number of patients did not qualify for chemoembolization. In addition, most patients are currently treated with cetuximab in 1st or 2nd line, which significantly limited the study size. 2. Table 2 shows the cox regression HR for predicting death. How did the authors differentiate the factors outside from the prior treatment of cetuximab. Is there any possibility that the nodule size diameter is the determinant of the survival rather than the prior treatment? As there are other factors are significantly different between those two groups, how to differentiate which factor is the determining one. For correct assessment, we again prepared the Cox proportional hazards model for all patients. ECOG performance status, liver involvement and low level of CEA was associated with better OS. Both, largest nodule size diameter and previous cetuximab treatment showed no significant impact on OS, but the difference for cetuximab treatment was close to significance HR 1.906 (0.977-3.716) p = 0.058. Larger group size could be helpful for accurate assessment, but the important information for us is the lack of difference in the effectiveness of chemoembolization in both groups, especially those treated with cetuximab. As suggested by other reviewers, we performed multivariate Cox proportional hazards models. 3. The title is a little bit of confusing, should at least include the test and the conclusion. We changed the title, as per recommendations. 4. Were the patients sequenced for mutation analysis? Is there any driver mutation recognized or could contribute to the survival difference between group A and B? We have added mutation types in table 1, there is a very large dominance of the KRAS mutation (EXon2). As mentioned in the discussion, there are studies which suggest that these mutations may impact survival. Despite large research groups, their results are often contradictory. Minors: 1. The figure 1 seems to not be included in any of the manuscript. Why is it in the introduction? What is it for? We removed figure 1. 2. The figure 2 might be better remake into a pie chart. We have made it into a pie chart. 3. Figure 3 is confusing. What is censored? „Censored” is the date of the last follow-up visit. It is too vague, need to at least be a 300 DPI figure and also need to include the p-value and sample size. Figure 3 has been revised as per your recommendations.

Reviewer 2 Report

The paper is interesting as it offers some information’s for a type of intervention

(chemoembolization) commonly used in the everyday practice for inoperable liver

metastatic lesions from colorectal cancer.

The method of study, the evaluation of the results and the discussion are on the way of

current scientific clinical research.

Author Response

Dear reviewer Thank you for your comments.

Reviewer 3 Report

This paper shows the results of TACE for unresectable colorectal liver metastasis, comparing a group of patients previously treated with Cetuximab versus some patients not treated before. 

Some major revisions:

1)Sample is small

2) The authors need to argue regarding the fact the PD and PR are similar but the OS is different. Which factors might have contributed? 

3)It needs to consider the number of TACE procedures performed for patients and to add this rate in table 1, table 2 and in the Cox regression analysis. Is possible, it would interesting to see a multivariate analysis. 

Minor revisions: the title is not well appropriate: why liver lesions? Not all patients of the study were previously treated with cetuximab

Author Response

Dear reviewer, thank you for your suggestions. Some major revisions: 1)Sample is small We agree, the study group could be larger. However, this is a single-center study with patients who have progression of disease after second and third lines of chemotherapy. Due to advanced disease and/or patients’ overall general coniditon, a large number of patients did not qualify for chemoembolization, which significantly limited the study size. In addition, most patients are currently treated with cetuximab in 1st or 2nd line. 2) The authors need to argue regarding the fact the PD and PR are similar but the OS is different. Which factors might have contributed? The OS correlation is close to the significance level of 0.05. Patients in group B were not treated with cetuximab due to the presence of KRAS or BRAS mutations. As mentioned in the discussion, there are conflicting studies which suggest that these mutations may impact survival. The important information for us is the similar effectiveness of chemoembolization in both groups. 3)It needs to consider the number of TACE procedures performed for patients and to add this rate in table 1, table 2 and in the Cox regression analysis. We have added this to tables and Cox’s model. Is possible, it would interesting to see a multivariate analysis. We performed multivariate Cox proportional hazard models. Minor revisions: the title is not well appropriate: why liver lesions? Not all patients of the study were previously treated with cetuximab We changed the title per your suggestions.